# VCAM-1 as a Biomarker of Endothelial Function among HIV-Infected Patients Receiving and Not Receiving Antiretroviral Therapy

**DOI:** 10.3390/v14030578

**Published:** 2022-03-11

**Authors:** Agnieszka Lembas, Katarzyna Zawartko, Mariusz Sapuła, Tomasz Mikuła, Joanna Kozłowska, Alicja Wiercińska-Drapało

**Affiliations:** 1Department of Infectious and Tropical Diseases and Hepatology, Medical University of Warsaw, Hospital for Infectious Diseases in Warsaw, 02-091 Warszawa, Poland; mariusz.sapula@wum.edu.pl (M.S.); tomasz.mikula6@wp.pl (T.M.); joanna.kozlowska@wum.edu.pl (J.K.); awiercinska@gmail.com (A.W.-D.); 2Students’ Science Society of the Department of Infectious and Tropical Diseases and Hepatology, Medical University of Warsaw, 02-091 Warszawa, Poland; katarzynalisik@gmail.com

**Keywords:** VCAM-1, HIV, cardiovascular, endothelium, marker

## Abstract

The Human Immunodeficiency Virus and retroviral therapy are both known risk factors for cardiovascular disease. It remains an open question whether HIV or ARV leads to increased arterial inflammation. The objective of this study was to investigate the changes in endothelial activation by measuring VCAM-1 levels among HIV-infected patients who were and were not treated with antiretroviral therapy. It is a retrospective study that included 68 HIV-infected patients, 23 of whom were never antiretroviral-treated, 15 who were ART-treated for no longer than a year, and 30 who were ART-treated for longer than a year. Blood samples were collected for biochemical analysis of the concentration of VCAM-1. The results show a statistically lower VCAM-1 level (*p* = 0.007) in patients treated with ART longer than a year (1442 ng/mL) in comparison to treatment-naïve patients (2392 ng/mL). The average VCAM-1 level in patients treated no longer than a year (1552 ng/mL) was also lower than in treatment-naïve patients, but with no statistical significance (*p* = 0.096). Long-term antiretroviral therapy was associated with the decline of VCAM-1 concentration. That may suggest the lowering of endothelial activation and the decreased risk of the development of cardiovascular disease among ARV-treated patients. However, VCAM-1 may not be a sufficient factor itself to assess this, since simultaneously there are a lot of well-known cardiovascular-adverse effects of ART.

## 1. Introduction

It is estimated that up to 38 million people in the world are infected with the Human Immunodeficiency Virus (HIV) [1]. Antiretroviral therapy is the best option for sustaining viral suppression and reducing HIV-related mortality. In Poland, over 12 thousand people living with HIV (PLHIV) are receiving antiretroviral treatment [2].

HIV is known to be a risk factor for cardiovascular disease (CVD) [3]. A cardiovascular disease is a group of circulatory system disorders whose underlying cause is most often atherosclerosis [4]. In general, population mortality rates from CVD are decreasing, whereas among HIV–infected patients they are even increasing [5]. Recent studies show that people living with HIV have a higher risk of CVD, particularly heart failure and stroke [6]. Even patients who receive effective therapy are still prone to increased arterial inflammation and impaired smooth muscle function [7], which promote atherosclerosis and plaque formation. These changes in blood vessels play a significant role in the development of cardiovascular disease [8]. HIV-infected patients undergoing antiretroviral therapy (ART) have a 50% to 200% higher risk of heart failure than the matched HIV-negative community [9]. These results remain the same even after the correction of standard risk factors such as age, gender, and smoking status [10,11].

Vascular cell adhesion molecule-1 (VCAM-1) is a protein that functions as a cell adhesion molecule [12]. When cytokines stimulate the endothelial cells, the VCAM-1 gene is expressed on both large and small blood vessels [13]. This phenomenon occurs during the inflammatory process. VCAM-1 triggers endothelial signaling through NADPH oxidase-generated reactive oxygen species. This leads to the opening of intercellular passageways for the migration of leukocytes [14]. However, quiescent endothelial cells do not induce VCAM-1 expression by themselves [15]. The response to cytokines and chemokines concerns not only endothelial cells but also cardiac cells, especially fibroblasts. They respond to several chemoattractants released during cardiac injury, which involves damage-associated molecular patterns. This, through many pathways such as PI3K/AKT or NF-KB, induces VCAM-1 expression, which therefore leads to the leukocytes’ recruitment [16].

Another important factor that initiates an inflammatory response is inflammasomes. The NLRP3 (NLR-family pyrin domain-containing protein 3) is a very characteristic inflammasome, whose assembly leads to caspase 1-dependent release of the cytokines IL-1β and IL-18 and pyroptosis, which is a programmed cell death. VCAM-1 is one of the triggers that induces the NLRP3 inflammasome. Vascular endothelium plays an important role in the regulation of inflammation progression, and therefore in cardiovascular implications, such as cardiovascular disease or metabolic syndrome. There are studies which investigate the influence of NLRP3 inflammasome-targeting drugs on endothelial dysfunction. This requires further studies; however, there are presumptions that the inhibition of the NLRP3 inflammasome could contribute to the improvement of endothelial functions [17].

Due to these properties, VCAM-1 is a diagnostic biomarker used in many clinical studies to estimate endothelial dysfunction, which is a risk factor for cardiovascular diseases [18].

This study aimed to assess the possible differences in endothelial activation in HIV-infected patients who were and were not treated with antiretroviral therapy. This was performed by the evaluation of the association between VCAM-1 concentration and the duration of antiretroviral therapy.

## 2. Materials and Methods

### 2.1. Patients

We studied the population of 68 adult patients. All data comes from the Department of Infectious and Tropical Diseases and Hepatology, the Medical University of Warsaw from the period of 2009–2014. The inclusion criteria were HIV infection, regardless of the stage of illness and time of treatment and age 18 or older. The exclusion criteria were the occurrence of ischaemic heart disease and undergoing lipid-lowering therapy.

### 2.2. Assessments

All patients underwent physical examination and laboratory testing. Blood samples were collected for total cholesterol, HDL-cholesterol, LDL-cholesterol, triglyceride level, CD4, CD8 count and percentage, and HIV viral load. VCAM-1 concentration was measured in serum with the usage of The Quantikine^®^ Human VCAM-1/CD106 Immunoassay. This method is a 2-h solid-phase ELISA that employs an enzyme-linked monoclonal antibody specific for human VCAM-1. Any VCAM-1 present in samples is sandwiched by that monoclonal antibody and the immobilized antibody. Then a substrate solution is added and color develops in proportion to the amount of VCAM-1. The lower limit of the VCAM-1/CD106 Immunoassay is 6.3 ng/mL [19].

### 2.3. Statistical Analysis

The ANOVA test was used to evaluate the difference in mean value among quantitative variables. The Tukey HSD test was used as a post hoc test for the assessment of statistical significance in variables among two groups of patients. The *p*-value was set at 0.05. All statistical analyses were performed using Python 3.7 software.

## 3. Results

### 3.1. Patients

Among 68 patients, 23 were ART-naïve (18 men, 5 women), 15 were receiving antiretroviral therapy shorter than a year (12 men, 3 women), and 30 were treated longer than a year (23 men, 7 women). The characteristics of the patients are presented in Table 1.

### 3.2. Antiretroviral Therapy

Table 2 presents the data concerning the composition of antiretroviral therapy used among examined patients. Of the group of 23 naïve patients, 9 were newly diagnosed with HIV, and 14 were not treated before.

### 3.3. Measurement of VCAM-1 Concentration in Healthy Volunteers

The mean VCAM-1 concentration among 36 healthy volunteers was 557 ng/mL. The results varied from 349 to 991 ng/mL and the standard deviation was 139.6 ng/mL [16]. No negative samples were obtained from healthy volunteers. Figure 1 presents the comparison of the results among the healthy volunteers and examined groups of patients.

### 3.4. VCAM-1 Concentration in Examined Groups of Patients

The ANOVA test was performed to assess the difference among VCAM-1 concentrations in patients with different durations of the antiretroviral therapy. Since it showed statistical significance (*p* = 0.008), we performed further analysis. There were no negative samples among the examined patients. Table 3 shows the results.

### 3.5. Coinfections

HIV infection was not the only chronic viral infection among our patients. In total, 31 out of 68 analyzed patients were coinfected. In total there were, 9 patients with the Hepatitis B Virus (HBV), 18 patients with the Hepatitis C Virus (HCV), 4 patients with both HBV and HCV. Therefore, we assessed the relationship between coinfection and VCAM-1 concentration. The results are shown in Table 4.

Since the *p*-value was <0.05, we performed the Tukey HSD test for further investigation and presented the results in Table 5.

We observed statistical significance in the VCAM-1 serum concentration among patients with HCV coinfection and patients who were not coinfected. The remaining coinfections were not statistically significant.

### 3.6. Age Correlation

The average age of patients from our three groups differed, especially between the naïve patients (the average of 34.3 years) and patients treated longer than a year (the average of 45.6 years). There was also a difference in the VCAM-1 level in those two groups, so we decided to examine the correlation between our patients’ age and the level of VCAM-1, which appeared to be statistically insignificant. We located the results in Table 6.

### 3.7. Smoking Cigarettes

We analyzed the habit of smoking among our patients. We classified smoking patients as those who were smoking at least 20 cigarettes per day for a minimum of 5 years. In treatment-naïve patients, 16 were smoking and 7 were not. Of patients treated for less than a year, nine were smoking and six were not. In addition, among patients receiving ART for longer than a year, 22 were smoking and 8 were not. Among 47 out of 68 analyzed patients declared tobacco use, which stated 69%. The results of the statistical analysis concerning smoking are presented in Table 7.

There was no significance in VCAM-1 concentration among patients who were or were not smoking cigarettes.

## 4. Discussion

Our findings point out that endothelial activation concerns many people living with HIV. This activation was indicated especially among people who were HIV-positive and were not receiving antiretroviral treatment. That may suggest that this group of people could be the most susceptible to the development of cardiovascular disease. Our results have been supported by studies that show that there exists an association between untreated HIV infection and an increased risk of CVD [20] and that a vascular inflammatory process that is reflected by a pattern of endothelial activation occurs through untreated HIV infection [21]. On the other hand, a higher risk of CVD and lipid metabolism disorders are the known adverse effects of ART [22]. At present, there are a lot of controversial debates about whether HIV-infected patients receiving antiretroviral therapy are more prone to developing CVD than those who are treatment-naïve [23]. Some studies show that patients receiving antiretroviral therapy (ART), including protease inhibitors (PI), more often develop lipodystrophy, dyslipidemia, direct mitochondrial DNA damage, and insulin resistance than HIV-positive individuals who are treatment-naïve [24]. The duration of exposure to ART as well as the drug class appeared to be the important factors in myocardial infarction. Among our patients, thirty-seven were treated with ART, including protease inhibitors, and we still observed the lowering of VCAM-1 serum concentration in comparison to treatment-naïve patients. In some studies, the levels of VCAM-1, which is the marker of endothelium activation, were lowered by short-term ART [25]. In contrast to those reports, the analysis of our patients does not show a statistically important difference between the group of patients treated less than a year and the group of non-treated patients. The reason for this situation is probably that the number of patients is different in both groups (15 patients treated for no longer than a year in comparison to 23 naïve patients). To summarize, it is evident that the initiation of effective ART ameliorates vascular inflammation but is not able to fully correct it. Research supports the hypothesis that the HIV viral load may directly result in an atherogenic milieu in untreated HIV infection [18], which can also be suggested in our analysis.

### 4.1. VCAM-1 Concentration According to Patients’ Age

Since our patients were at different ages, it is essential to consider this. Chronic inflammation, which comprises the endothelium, is associated with HIV infection [26] and the natural aging process without HIV infection [27]. However, endothelial dysfunction occurs earlier and is accelerated in HIV-positive subjects [28]. The correlation between our patients’ age and the level of VCAM-1 concentration appeared to be statistically insignificant. Other studies seem to support these results by showing that there is no elevated frailty caused by the HIV-associated inflammation that could be raveled in similarly aged uninfected individuals [29].

### 4.2. Smoking Cigarettes and VCAM-1 Concentration

Another important subject is tobacco usage, since 69% of our patients were smoking at least 20 cigarettes per day for a minimum of 5 years. Globally, smoking is more common among people who are HIV-infected than in the general population [30]. However, we did not observe a statistically significant difference in VCAM-1 serum concentration among patients who were and were not smokers. That could be the result of the small population of non-smoking patients in our study. Studies show that smoking is associated with an elevated cardiovascular risk, including coronary artery disease, peripheral vascular disease, ischaemic heart disease, atherosclerosis, myocardial infarction, and stroke [31]. Tobacco use, as well as HIV infection itself, were associated with increased endothelial biomarker levels [32].

### 4.3. Impact of Coinfections on VCAM-1 Concentration

HIV, HBV, and HCV are the most common chronic viral infections documented worldwide [33]. The causes of this phenomenon have similar ways of spreading, which are blood and blood products, sharing of needles to inject drugs, and sexual activity [34]. In our patients, co-infections were mostly occurring in the group of individuals that were ART-treated for longer than a year (16 patients).

Chronic Hepatitis C (CHC) patients are more likely to suffer from both liver disease and cardiovascular disease (CVD). Studies have shown a higher spread of type 2 diabetes mellitus (DM), insulin resistance, and hepatic steatosis, which are known CVD risk factors in HCV-positive patients compared to uninfected individuals [35]. Furthermore, recent studies have shown that HCV infection is a direct risk factor for subclinical and clinical cardiovascular disease (CVD) [36] and is directly connected to arteriosclerosis. HCV RNA sequences have been found in the plaque tissues of patients who underwent carotid revascularization. It shows that HCV RNA sequences seem to play a local effect on the endothelium [37], which can be the reason for increased levels of VCAM-1 in patients with liver diseases such as liver cirrhosis or chronic hepatitis C (CHC). Among our patients, there were 18 individuals infected with both HIV and HCV, and the level of VCAM-1 concentration was significantly increased in this group of patients. Some studies have corresponded with a slight increase in CVD among subjects with HIV-HCV co-infection in contrast to those without HCV co-infection [38], and our study supports this thesis.

On the other hand, HBV was not related to the time of CVD occurrence [38]. The natural mileage of HBV infection is meaningfully altered by HIV co-infection, as it is more aggressive with higher HBV DNA levels and lower inflammatory activity [39]. We also did not observe a significantly higher concentration of VCAM-1 among patients with HBV co-infection. It is also known that HIV, HBV, and HCV co-infection are connected with liver-related deaths [40]. Only four of our patients were co-infected with both HCV and HBV, and therefore we cannot conclude about the CVD occurrence in this population.

### 4.4. Dyslipidemia and Endothelial Dysfunction

Studies suggest that there is a higher prevalence of dyslipidemia among HIV-infected patients [41]. Dyslipidemia concerns both ART-treated and non-treated patients and is associated with a higher risk of stroke and myocardial infarction [10]. Since it is closely connected to CVD, we decided to evaluate the differences in total cholesterol, triglycerides, HDL-cholesterol, LDL-cholesterol among patients who are treatment-naïve, and ART-treated for shorter and longer than a year. Both total cholesterol and LDL-cholesterol levels appeared significantly higher among patients undergoing antiretroviral therapy than in treatment-naïve individuals, whereas HDL-cholesterol and triglycerides were statistically insignificant. However, despite the elevation of the concentration of total cholesterol and LDL-cholesterol, the VCAM-1 concentration was significantly lower among the treated patients.

### 4.5. Severity of HIV Infection

Our study showed statistical significance in CD4, CD8 count, and viral load among the three groups of patients. These results are predictable since effective antiretroviral therapy leads to an increase in CD4 count and a decrease in VL [42]. However, studies discussing the topics of CD4 count, viral load, and endothelial dysfunction are not unambiguous. Some researchers show that more viral copies may have a negative impact on the endothelium [43]. Others claim that neither CD4 count, nor HIV viral load, are the predictors of endothelial dysfunction [44].

### 4.6. VCAM-1 Targeting Molecules

Evidence suggests that VCAM-1 is associated with multiple disorders such as cardiovascular disease, cancer, rheumatoid arthritis, and asthma. There are many molecules that trigger the expression of VCAM-1. One of the most characteristic VCAM-1 inducing cytokines is TNF-α [45]. It is a member of the TNF/TNFR cytokine superfamily, which is involved in the maintenance of the immune system but also plays an important role in chronic inflammation [46]. Studies also point out that alpha D beta 2, a member of beta 2 integrins, can support eosinophil adhesion to VCAM-1. Moreover, alpha D beta 2 binds to VCAM-1 and can also support lymphoid cell adhesion to VCAM-1 [47]. One study developed the theranostic nanocarriers decorated with VCAM-1 antibodies that seem to localize the endothelial senescence and prevent pro-senescent endothelial responses [48].

Nutraceuticals, which are dietary supplements, are currently gaining attention due to their therapeutic potential [49]. There is research suggesting that some nutraceuticals could improve vascular function. There is a study that tested the use of hyaluronic acid hydrogel of Quercetin on human thyroid cancer cells. Quercetin showed an anti-inflammatory effect via a CD44-dependent interaction with thyroid cancer cells [50]. Since CD44 is known to bind VCAM-1, which decreases tumor growth [51], it is possible that Quercetin could reduce the expression of VCAM-1 and therefore endothelial activation.

### 4.7. The Role of Hyaluronic Acid in the Imrpovement of Endothelial Function

Hyaluronic acid is a natural polysaccharide that commonly occurs in human bodies. Moreover, it is being used as a pharmaceutical, mostly in ophthalmology. Hyaluronic acid can be cross-linked or conjugated with multiple biomacromolecules. Studies report the importance of interactions of hyaluronic acid with the CD44 receptor in pathological processes such as cancer. Research shows that controlled release of proteins and pharmaceuticals from hyaluronic acid resulted in many benefits in cancer treatment, which included, for example, an enhanced therapeutic effect with minimum toxicity [52]. Another study showed that cross-linked hyaluronic acid sub-micron particles can recognize cancer tissue and can be used to deliver bioactives in a specific and controlled manner to cancerous tissue [53]. The CD44 receptor is crucial in HIV infection. It enhances the infection in CD4(+) T cells. Studies are being developed to assess the role of hyaluronic acid in the mucosal transmission of HIV. Hyaluronic acid seems to reduce HIV infection during the interaction of HIV with CD4(+) in a CD44-dependent manner. It could be relevant to HIV mucosal transmission in general [54]. Since the CD44 receptor also binds VCAM-1, it could also be helpful in reducing endothelial activation. The role of hyaluronic acid in this process is not clear, and it remains an open topic for further studies.

### 4.8. Limitations of the Study

The main limitation of our study was the small study population. Moreover, we did not perform the calculation and justification of the sample size selected. All of our patients were HIV-positive, since we were not able to compare our results to HIV-negative individuals.

## 5. Conclusions

In our study, we observed that the concentration of VCAM-1, which is the marker of endothelial activation, was statistically lower in HIV-infected patients on antiretroviral therapy for longer than a year than in HIV-infected patients who were not receiving the therapy. That means that effective antiretroviral therapy may potentially inhibit the influence of the virus on the endothelium. This leads us to the conclusion that it is possible that long-term antiretroviral therapy may be associated with a lower risk of cardiovascular disease than untreated HIV infection. However, the lowering of VCAM-1 concentration may not be a sufficient factor to assess the improvement of endothelial function. It requires further studies since, simultaneously, there are a lot of well-known cardiovascular adverse effects of ART, including dyslipidemia and hyperglycemia.

## 6. Patents

This section is not mandatory but may be added if there are patents resulting from the work reported in this manuscript.

## Figures and Tables

**Figure 1 viruses-14-00578-f001:**
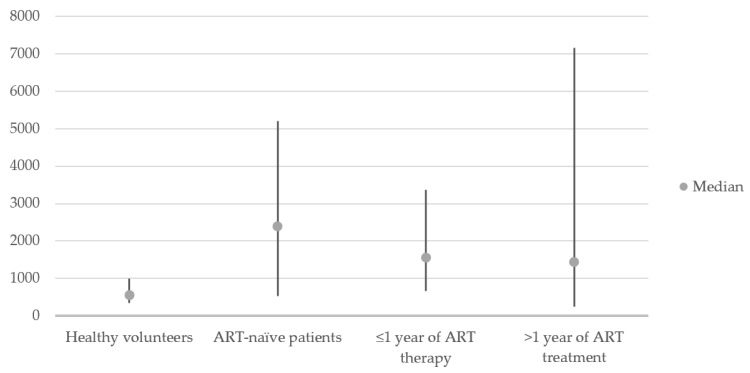
Distribution of VCAM-1 concentrations in healthy volunteers and examined groups of patients.

**Table 1 viruses-14-00578-t001:** Basic characteristics of the patients.

Characteristics of the Patients	ART-Naïve Patients	≤1 Year of ART Therapy	>1 Year of ART Treatment	*p*
Age (years)	34.3 (25–49)	34.4 (21–74)	45.6 (30–70)	0.000
VCAM-1 (ng/mL)	2392 (534–5198)	1552 (662–3364)	1442 (246–7166)	0.008
Total cholesterol (mmol/L)	3.66 (1.28–4.94)	4.36 (2.74–7.18)	4.42 (2.17–6.25)	0.032
LDL-cholesterol (mmol/L)	1.94 (0.47–4.48)	2.53 (1.36–4.94)	2.53 (1.22–4.44)	0.041
HDL-cholesterol (mmol/L)	1.09 (0.4–2.1)	1.22 (0.68–1.9)	1.39 (0.32–2.9)	0.108
Triglyceride (mmol/L)	1.61 (0.82–2.9)	1.61 (0.93–2.57)	1.93 (0.88–5.29)	0.305
CD4 (cells/µL)	212 (6–482)	282 (27–704)	413 (103–791)	0.003
CD4 (%)	24.6 (3–58)	25.5 (6–56)	34 (11–67)	0.056
CD8 (cells/µL)	537 (71–1391)	831 (65–1770)	921 (78–2666)	0.041
CD8 (%)	73.7 (41–92)	71.1 (45–89)	67 (40–91)	0.233
CD4:CD8	0.398 (0.04–0.9)	0.399 (0.07–1.26)	0.597 (0.09–1.7)	0.117
Viral load (copies/mL)	901,160 (0–10,000,000)	83,557 (0–746,695)	11,499 (0–226,006)	0.031
Co-infections	2 patients–HBV 5 patients–HCV 2 patients–HBV/HCV	2 patients–HBV 2 patients–HCV 2 patients–HBV/HCV	5 patients–HBV 11 patients-HCV	0.047
Smoking cigarettes	16	9	22	0.185
Length of therapy (weeks)	0	3–52	76–988	0.000
Average length of therapy (weeks)	0	21.6	300.4	0.000
Median length of therapy (weeks)	0	20	222.5	0.000

**Table 2 viruses-14-00578-t002:** Antiretroviral therapy among patients.

Applied Antiretroviral Therapy	(*n*)
Nucleoside Reverse Transcriptase Inhibitors (NRTI)	92
Protease Inhibitors (PI)	39 ^1^
Non-nucleoside Reverse Transcriptase Inhibitors (NNRTI)	9
Integrase Inhibitors (II)	2

^1^ Four without ritonavir as a booster.

**Table 3 viruses-14-00578-t003:** Post hoc tests assessing VCAM-1 concentrations and length of antiretroviral therapy using Tukey HSD test.

Compared Groups -Length of the Antiretroviral Therapy (Years)	Average Differential in VCAM-1 Concentration (ng/mL)	*p*
Naïve vs. treated ≤1 year	840 (−180–1800)	0.096
≤1 year vs. treated >1 year	200 (−740–1120)	0.871
Naïve vs. treated >1 year	1040 (240–1840)	0.007

**Table 4 viruses-14-00578-t004:** The results of the ANOVA test assessing coinfections and VCAM-1 concentration.

	No Coinfection	HCV Coinfection	HBV Coinfection	HCV and HBV Coinfections	*p*
VCAM-1 concentration	1453.8 (246–3826)	2497.8 (628–7166)	2265.4 (664–5198)	2207.6 (1122–3130)	0.047

**Table 5 viruses-14-00578-t005:** The results of the Tukey HSD test comparing coinfections and VCAM-1 concentration.

Compared Groups of Patients-Coinfections	Average Differential in VCAM-1 Concentration (ng/mL)	*p*
No coinfections vs. HCV coinfection	1044 (9.2–2078.8)	0.047
No coinfections vs. HBV coinfection	811.6 (−503.2–2126.4)	0.371
No coinfections vs. HCV and HBV coinfections	753.8 (−1110.6–2618)	0.689
HCV coinfection vs. HBV coinfection	−232.4 (−1694.4–1229.6)	0.9
HCV coinfection vs. HCV and HBV coinfections	−290.2 (−2261.2–1680.6)	0.9
HBV coinfection vs. HCV and HBV coinfections	−57.8 (−2189–2073.4)	0.9

**Table 6 viruses-14-00578-t006:** The correlation between patients’ age and VCAM-1 level.

The Group of Patients–Length of Antiretroviral Therapy	r-Value	*p*-Value
All patients	−0.14	0.244
ARV-naive patients	0.20	0.334
ARV ≤ 1 year	0.06	0.818
ARV > 1 year	−0.08	0.672

**Table 7 viruses-14-00578-t007:** The statistical significance of smoking and VCAM-1 concentration.

	VCAM-1 Concentration in Smoking Patients	VCAM-1 Concentration in Non-Smoking Patients	*p*
VCAM-1 concentration	2017.2 (246–7166)	1529.2 (528–5198)	0.185

## Data Availability

Not applicable.

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
