# Peer review of "VCAM-1 as a Biomarker of Endothelial Function among HIV-Infected Patients Receiving and Not Receiving Antiretroviral Therapy"

_viruses, 2022, doi:10.3390/v14030578_

Round 1

Reviewer 1 Report

This is a cross-sectional study of VCAM-1 levels in persons with HIV +/- ART in Poland.

The study population is modest in size.  This is a major limitation given the variable within the population (e.g., aging, HBV and HCV co-infections, distinct ART regimens, smoking).  I would strongly advise increasing the population size to permit more meaningful conclusions.

What is the lower limit of the VCAM-1/CD106 immunoassay?  Were negative samples tested a second time to confirm that finding?

Are the P values shown in Table 1 comparing all 3 groups to one another or two of the groups (if so, which ones)?

Table 2 is confusing.  N is shown but 92 for NRTI is higher than the stated number of people in the entire study!

A figure with all VCAM-1 levels for the 3 HIV study groups and the healthy controls would be helpful in showing the distribution of VCAM-1 levels.  This variability is somewhat obscured by showing means.

Are the VCAM-1 data normally distributed in all study groups or would a comparison of medians be more appropriate?

Author Response

Point 1. The study population is modest in size.  This is a major limitation given the variable within the population (e.g., aging, HBV and HCV co-infections, distinct ART regimens, smoking).  I would strongly advise increasing the population size to permit more meaningful conclusions.

Response 1. Thank You very much for Your review. We agree that increasing the population would strongly contibute to the value of our study. Unfortunately the study was already finished, we no longer have the VCAM-1 agents at our disposal and therefore it is not possible to increase the study population.

Point 2. What is the lower limit of the VCAM-1/CD106 immunoassay?  Were negative samples tested a second time to confirm that finding?

Responce 2. The lower limit of the VCAM-1/CD106 Immunoassay is 6.3 ng/mL. There were no negative samples among our patients and also among healthy volunteers. The proper information is now included in the text.

Point 3. Are the P values shown in Table 1 comparing all 3 groups to one another or two of the groups (if so, which ones)?

Rensponce 3. The P values in Table 1 are comparing all 3 groups to one another.

Point 4. Table 2 is confusing.  N is shown but 92 for NRTI is higher than the stated number of people in the entire study!

Response 4. The Table was ment to show the usage of pharmaceuticals from the different groups of ARV therapy among our patients. 92 NRTI results from the fact that some patients scheme of therapy included more than one NRTI drug.

Point 5. A figure with all VCAM-1 levels for the 3 HIV study groups and the healthy controls would be helpful in showing the distribution of VCAM-1 levels.  This variability is somewhat obscured by showing means.

Response 5. The figure including distibution of VCAM-1 levels, as well as means, was made and it is now included in the manuscript.

Point 6. Are the VCAM-1 data normally distributed in all study groups or would a comparison of medians be more appropriate?

Response 6. The VCAM-1 data is normally distibuted in all study groups.

Reviewer 2 Report

The manuscript titled "VCAM-1 as a biomarker of endothelial function among HIV-infected patients receiving and not receiving antiretroviral therapy" is a very interesting original article describing the role of VCAM1 as biomarker of vasculotoxicity induced by a viral disease. The article is well orgainzed, the overall structure is of good quality; methods are clear and references are updated. However, manuscript should improved in several parts:

1) Introduction should be improved: authors shuld explain the role of VCAM-1 in endothelial cells and also in cardiac cells. A brief description of the intracellular pathways should be made; what about the NLRP3 inflammasome induced by VCAM1? how NLRP3 inhibition could improve endothelial functions? 

2) Authors should improve the discussion with an overall description of VCAM1-targeting molecules and also nutraceuticals that could improve vascular functions. For example, quercetin is able to reduce several cytokines involved in VCAM-1 pathways. Authors should explain how some nutraceuticals could improve endothelial functions ( you can cite 10.1002/jcp.25283 )

3)Authors should explain how hyaluronic acid at low or high molecular weight could be a suitable pharmaceutical formulation that could improve endothelial functions through VCAM-1 pathways.  A preclinical and clinical description of the therapeutic potential of hyaluronic acid-based formulations should be made both for HIC patients and cancer patiens as well. ( you can cite 10.1007/s10856-013-4895-4 and doi: 10.1038/icb.2014.50 )

Author Response

Point 1. Introduction should be improved: authors shuld explain the role of VCAM-1 in endothelial cells and also in cardiac cells. A brief description of the intracellular pathways should be made; what about the NLRP3 inflammasome induced by VCAM1? how NLRP3 inhibition could improve endothelial functions?

Response 1. Thank You very much for Your review. The introduction was developed and it now includes the role of VCAM-1 in endothelial and cancer cells. The NLRP3 inflammasome role in endothelial function was also mentioned.

Point 2. Authors should improve the discussion with an overall description of VCAM1-targeting molecules and also nutraceuticals that could improve vascular functions. For example, quercetin is able to reduce several cytokines involved in VCAM-1 pathways. Authors should explain how some nutraceuticals could improve endothelial functions ( you can cite 10.1002/jcp.25283 )

Response 2. The discussion was improved. VCAM-1 targeting molecules and the potential role of nutriceuticals in the improvement of vascular functions were included. The proposed citation is now included in the text.

Point 3. Authors should explain how hyaluronic acid at low or high molecular weight could be a suitable pharmaceutical formulation that could improve endothelial functions through VCAM-1 pathways.  A preclinical and clinical description of the therapeutic potential of hyaluronic acid-based formulations should be made both for HIC patients and cancer patiens as well. ( you can cite 10.1007/s10856-013-4895-4 and doi: 10.1038/icb.2014.50 )

Response 3. The potential role of hyaluronic acid in the improvement of endothelial function and also in HIV infection in general was included in the manuscript. The clinical description of therapeutic potential of hyaluronic acid was made also for cancer patients. The proposed citations were included in the text.

Round 2

Reviewer 1 Report

previous comments have been addressed appropriately